# Manufacture and Performance of Welds in Creep Strength Enhanced Ferritic Steels

**DOI:** 10.3390/ma12142257

**Published:** 2019-07-13

**Authors:** Jonathan Parker, John Siefert

**Affiliations:** Electric Power Research Institute, Charlotte, NC 28262, USA

**Keywords:** steel, weld, high temperature, creep, fracture, advanced methods

## Abstract

Welding is a vital process required in the fabrication of ‘fracture critical’ components which operate under creep conditions. However, often the procedures used are based on ‘least initial cost’. Thus, it is not surprising that in many high energy applications, welds are the weakest link, i.e., damage is first found at welds. In the worst case, weld cracks reported have had catastrophic consequences. Comprehensive Electric Power Research Institute (EPRI) research has identified and quantified the factors affecting the high temperature performance of advanced steels working under creep conditions. This knowledge has then been used to underpin recommendations for improved fabrication and control of creep strength enhanced ferritic steel components. This review paper reports background from this work. The main body of the review summarizes the evidence used to establish a ‘well engineered’ practice for the manufacture of welds in tempered martensitic steels. Many of these alternative methods can be applied in repair applications without the need for post-weld heat treatment. This seminal work thus offers major benefits to all stakeholders in the global energy sector.

## 1. Introduction

EPRI is a not for profit organization that has been providing independent technical support to global stakeholders in the electricity supply industry for over 40 years. Within EPRI’s generation sector, a key research imperative is knowledge creation and technology transfer linked to the reliable, safe, and economically flexible operation of power plants. Collaborative achievements have included contributions to the development of databases of key properties for high temperature alloys, publication of recommended guidelines for design and fabrication as well as compiling case studies of in-service problems and facilitating expert root cause assessment. Technology transfer has ensured that lessons learned can be used to establish best practices; these activities include annual workshops, the publication of summary documents and identification of additional research. In all cases, excellence in science and engineering is necessary to underpin technology which will help to meet challenges associated with the safe and reliable operation of a plant.

The present paper reviews key findings from over 10 years of collaborative research which was carried out to identify and quantify the factors affecting the high temperature behaviour of tempered martensitic steels. In particular, selected information is provided from a series of EPRI-coordinated, industry-sponsored projects on the manufacture and performance of welds made in 9%Cr creep strength-enhanced ferritic (CSEF) steels. The initial work in this area provided the basis for a meaningful asset management strategy for Grade 91 steel components. This seminal project involved more than 40 participants providing over $4 million of industry funding. The learnings and findings were realized with direct input and perspective from stakeholders representing the entire electricity supply chain. The knowledge created established that in addition to operating variables such as stress state and temperature, the high temperature behaviour of CSEF steels was dependent on a Metallurgical Risk Factor. Meaningful assessment of component creep behaviour could thus only be achieved by integrating information from steel making and fabrication with microstructure and operating conditions, with particular emphasis on complex transient loading.

A key outcome from this research was that for metallurgically complex steels such as CSEFs, it is vital that research programmes should establish high temperature performance on steel sections which are carefully chosen and well characterized [1]. Thus, it is important that the full pedigree of the steel heat or cast selected for a research project is known or checked. For example, when research is assessing the creep behaviour of welds, the welds must be made in the same base substrate so that results can be directly compared. Similarly, it should not be assumed that the behaviour of welds made in plate will exhibit the same performance trends as welds made in tubes or pipes.

Achievements directly linked to factors controlling the high temperature performance of welds in CSEF steels are reviewed in this paper with detailed background provided in the original reports and papers [1,2,3]. The primary areas covered are summarized below:Detailed research examining the microstructure of tempered martensitic ferritic steels has provided key information concerning the rate controlling damage mechanisms for different creep conditions. Initially sections from early in life component failures were examined to establish damage patterns and to highlight the critical evidence, see for example Figure 1. The root cause analyses performed [4] have established that the inherent cavity susceptibility of the Grade 91 base steel is a key factor which is directly linked to damage development in the heat affected zone (HAZ) of welds. This susceptibility also influences behaviour under cyclic creep conditions (often described as creep fatigue).The microstructure in the heat-affected zone (HAZ) of welds made in 9% chromium martensitic steels is complex. Systematic investigations using advanced methods have defined the different regions of the microstructure across the HAZ as functions of the precipitate condition and the welding process [5,6]. These regions are categorized in terms of the influence of the thermal cycles on the sub-structural parameters controlling creep deformation and fracture. Results from cross weld feature tests then provided direct evidence to rationalize the apparently conflicting published evidence regarding nucleation, growth and linkup of creep cavities in the HAZ of welds.Continuum damage mechanics (CDM) based methods have been available for some time. However, recent research has focused on technically relevant descriptions to evaluate both when a component will fail and also how it will fail [7]. For many operators and end users, there is a significant benefit to selecting component design and manufacturing methods which are inherently Damage Tolerant, i.e., there is a long period between crack initiation and fracture. Application of CDM has shown that the inherent advantages of reducing the risks of catastrophic fracture do not require significant greater financial investment [8,9]. These benefits can instead be realized by the application of sound engineering approaches to weld design and manufacture.A clear technical route has established the improved design and fabrication for well-engineered welds in CSEF steels. Applying engineering best practice to welds for high energy applications is always of value. However, particular benefit has been found with establishing three alternative options for repair of Grade 91 steel components [10]. In particular, research, conducted over the last five years, has identified that repair welds in CSEF steels can be made without the need for post weld heat treatment. Indeed, detailed tracking of real components showed that badly controlled PWHT actually resulted in problems. Expert review of the EPRI weld repair methods has culminated in the acceptance within the National Board Inspection Code, USA, of Welding Method 6 for tubes and Welding Supplement 8 for thick section parts [11].

It is apparent that the uncertainty regarding component performance is in part a consequence of inadequacies during the design and fabrication. Sometimes less than optimal approaches are justified on the basis of least cost. Failure to use best practice specification, design and manufacturing methods may also be the result of a lack of knowledge or lack of control or both. In all cases, excessive variability in the condition of a component or structure at the start of service life results in major downstream uncertainties which may include problems with fractures and forced outages. With particular reference to the fabrication and high temperature performance of CSEF steels, it is apparent that the EPRI knowledge base offers technical information and guidance to minimize variability in behaviour and thus greatly simplifies asset management [12]. The information described in this review paper should be used by designers, manufactures and end users to help to meet the challenges of economic, safe and reliable operation of high energy components.

## 2. Metallurgical Risk Factor

The high temperature performance of CSEF steels is dependent on the details of steel making, steel processing, composition, and heat treatment [1]. Moreover, it is not possible to simply ‘recover’ the performance of a poorly made steel batch by application of a subsequent heat treatment. Thus, the final quality or the steel to be used starts with high quality in composition control, steel making, poring etc. While a steel section with a poor microstructure cannot be easily fixed by heat treatment, it is now very well established that the poor control of heat treatment can lead to different problems. Thus, for example, when irregularities of heat treatment result in a ferrite microstructure (rather than the martensitic structure expected), the creep strength is significantly reduced compared to the creep strength of tempered martensite.

EPRI recommendations clearly advocate performing detailed metallographic characterization and documentation of the damage present after creep testing. This assessment is particularly important for tests of long duration since these results are generally considered most representative of behaviour of components in service. In CSEF steels these long-term tests generally fracture with very little reduction of area. This creep brittle behaviour is a consequence of the formation and growth of cavities and the subsequent formation of micro and macro cracks. Careful examination using modern techniques has shown that creep cavities are formed both on prior austenite grain boundaries and other features in the martensitic microstructure such as at lath boundaries and at inclusions. The diversity of the microstructural sites which develop cavities is illustrated in Figure 2. It appears that the formation of voids at these diverse sites makes the formation of micro cracks more difficult.

The diversity of void nucleation sites creates a challenge to tracking in-service component damage using traditional inspection methods. While the details of the number of voids formed, and the tendency for reductions in strain to fracture, is different for the different CSEF steels, research to date [1] shows that void nucleation is related to the presence of trace elements and hard nonmetallic inclusions. In Figure 3 detailed characterization of the inclusions present has been performed. The steel identified as Barrel 2 has a relatively high inclusion number density and exhibits low creep ductility. In contrast, the steel identified as Tee piece 1 has a relatively low number density of inclusions and exhibits a high creep ductility.

An example of the link between the nucleation of a creep void, the presence of inclusions and segregation of trace elements is illustrated by the analysis results shown in Figure 4. The analysis reveals that high levels of copper are found associated with the creep void. Copper in this location can act to promote cavitation by reducing the local surface energy. A further key factor in determining whether inclusions aid the nucleation of voids is the particle size. Thus, only inclusions of a sufficient size (the critical inclusion size is directly linked to the creep stress) will be thermodynamically stable and thus able to act directly as nucleation sites.

It is clear from root cause analyses of Grade 91 steel components [1,4] that steel composition and processing variables are linked to low creep ductility in base metal and creep cracking in weld HAZs. The observed in-service component damage cannot be simply explained as a ‘one off’ anomaly. Examination of failures in several Grade 91 welded components [13,14] indicates a trend where an increasing number of failures of Grade 91 welds are occurring in times below that expected based on simple design rules. The trend in the reduced creep ductility in Grade 91 steel and the link to very low HAZ creep life may be a characteristic of a significant number of components which entered service with compositions which show a high susceptibility for cavity formation. It should be emphasized that the poor creep performance of weld HAZs is a key reason for the introduction of weld efficiency factors. These factors were aimed at reducing the risk of in-service damage by increasing the component thickness. Further, reductions in weld creep performance or indeed greater uncertainty over long term performance would be expected to lead to increases in recommended weld strength reduction factors (WSRF) or weld-efficiency factors. The evidence from EPRI research shows using Grade 91 steel with low densities of inclusions and controlled levels of deleterious trace elements should significantly reduce the risk of creep cracking associated with weldments. Selection of improved base material would provide significant benefit for high energy systems.

In order to properly assess metallurgical risk, it is necessary to obtain a full chemical composition for Grade 91 steel base material. As highlighted previously [1,15,16], there is concern that the influence of tramp elements such as As, S, Sn, Sb and Cu has been underappreciated and that these elements are playing a role in the reduction in creep ductility in martensitic CSEF steels. In general, the analysis of elements can be grouped into two sets: elements required by common specifications for Grade 91 steel (14 total elements) [1] and elements for informational purposes (typically 10+ additional elements). The following approaches are typically used to determine the composition of each of the elements in EPRI research. Inductively coupled plasma optical emission spectrometry (ICP-OES) was utilized to determine the values for: Al, B, Ca, Co, Cr, Cu, La, Mn, Mo, Nb, Ni, P, Si, Ta, Ti, V, W, Zr. Inductively coupled plasma mass spectrometry (ICP-MS) was used to determine the amounts of As, Bi, Pb, Sb, Sn. Finally, combustion was necessary to determine the C, S levels while insert gas fusion (IGF) to assess the amount of O and N in the steel. It should be noted that, to ensure that sufficient information is provided for each of the requested elements, the number of required significant digits should also be specified in any specification to the laboratory performing the analysis.

Discussions of industry experience in general, and the EPRI recommendations in particular, have now resulted in agreement within ASME that for Grade 91 steels there should be a Type I and a Type II designation [16]. It has been recommended that the steel made with greater controls designated Type II, will have higher Allowable Design stress values [16]. This recommendation is reflected in the ASME Code Case 2864, Table 1 and associated documentation. Similar specifications have been approved by ASTM for the component specific requirements. Discussions continue about the influence of trace elements on creep performance of other CSEF steels. However, the position recommended by EPRI is clear. In all cases, purchase of CSEF steel components should recognize the complexity of the metallurgy, the need for care to achieve excellent creep performance without excessive variability. Thus, well controlled composition and fabrication should be mandated.

## 3. Weld Manufacture and HAZ Microstructural Characterization

Proper characterization of the microstructure in metallurgically complex steels is complicated by the fact that most of the features which define creep strength and ductility cannot be resolved or studied using optical metallography. Moreover, the diversity of thermal cycles experienced by multi-pass fusion welds further complicates meaningful characterization because without care there is a very significant spatial variation throughout the weld and HAZ.

The preferred approach to overcome the problems of relevant examination and recording meaningful information, it is usually necessary to balance the results from macro-, micro- and nano- evaluation with appropriate analysis. Thus, it is important to take an overall view to microstructure and then increase detail based on the information recorded. In view of the challenges associated with nano level examination this level of detail can only be performed selectively. It is thus vital to make sure that the methods used for this selection are rigorous.

This section summarizes information regarding EPRI recommended approaches [17] for this characterization for welds manufactured in CSEF steels. It should be emphasized that in all cases the procedures used have been validated and checked against calibrated sections.

### 3.1. Weld Manufacture

EPRI has published a series of documents which describe procedures for weld manufacture. The outline below is included here because it is critical to research projects that the base sections used is well pedigreed and the subsequent welding is performed in a controlled manner. Lack of control of fusion welding will lead to a very wide range of local temperature cycles and therefore microstructures in any weld.

In thick section components such as pipes EPRI has typically used a machined U-groove with a 15° bevel and using best practice guidance for the shielded metal arc welding (SMAW) process as detailed previously [10]. For Grade 91 steels the process included a minimum preheat temperature of 150 °C (300 °F), a maximum interpass temperature of 315 °C (600 °F), stringer beads only, and removal of slag after each weld layer through light grinding. The filler material used to make the weldments was consistent with an American Welding Society (AWS) type E9015-B9 filler material. Stipulating stringer beads without weaving and using only 3.2 mm (0.125 inches) diameter electrodes limited the variability in the heat input. The completed weldment including a macro sample, documented fill sequence, and the recorded data for amperage, voltage, travel speed, and interpass temperature are provided elsewhere [10].

Following welding, the weldment was allowed to cool to room temperature. In some cases, EPRI research into weld repair investigated the performance of welds without post weld heat treatment (PWHT). In other cases, and for the weld shown in Figure 5, a relatively low temperature of PWHT, namely at 675 °C (1250 °F) held for 2 h, was used. Full details of the development and testing of welds made using alternative weld and PWHT procedures have been reported [10].

The weld macro section shown in Figure 5A demonstrates that the appearance of the weld was consistent with the expected bead sequence. This is shown in Figure 5B. Thus, it was clear that the requirements of the procedure had been followed. No obvious regions of inhomogeneity in the weld macrostructure were identified. Because of the limitations of optical metallography regarding defining microstructural differences, to provide visual images of the macroscopic bead size and shape and the overall pattern of structure EPRI has developed procedures for macro-analysis which includes hardness mapping [17]. The equipment utilized for the hardness mapping characterization was a LECO Automatic Hardness Tester, Model AMH-43. Hardness mapping was conducted so that the requirements in both ASTM E384-11 (ASTM 2011) and ISO 6507 (ISO 2005) were met. One of the key requirements in these two standards is that for a given hardness load (e.g., for this study 0.5 kgf), the indents should be at least 2.5 d apart (where d = mean diagonal distance of the measured Vickers indent in the material being examined).

To ensure that sufficient resolution in the data was obtained, i.e., to achieve enough indents in the heat affected zone, a very large area around the weld fusion boundary was analyzed. Thus, in contrast to simple hardness line scans carried out in some traditional studies, in the present research representative portions of the base metal, heat affected zone (HAZ) and deposited filler metal were captured in the hardness map. This approach resulted in a final hardness map size that was 25 mm × 25 mm and included a total of 10,000 indents. The location examined is shown as the highlighted box in Figure 6A with the hardness map produced shown in Figure 6B.

It should be noted that the regions of highest hardness in the weld were recorded at, or close to, the centre of the individual beads. The pattern exhibited by these regions shows the uniformity of the beads deposited. The highest hardness recorded were values that are <350 HV 0.5. This observed maximum value is higher than for typical weldment manufacturing where the hardness values are reduced to ~<300 HV 0.5. The difference is attributed to the specified PWHT namely (675 °C for 2 h), which is consistent with the recognized minimum in the National Board Inspection Code Part 3 Repairs and Alterations Supplement 8 (NBIC 2017). This Code provides post construction repair requirements for power generation components and materials including Grade 91 steel and is commonly utilized in North America and in some southern European countries.

It should also be noted that recently ASME B&PV Code Section I reduced the specified minimum PWHT requirements for Grade 91 steel type welds in new construction. The stipulated minimum conditions are 705 °C (1300 °F) for weld thickness > 13 mm (0.50 inches) and 675 °C (1250 °F) for weld thickness ≤ 13 mm (0.50 inches) [18].

#### Heat Affected Zone Microstructure

Historically, the microstructures in Low Alloy Steel welds have been classified based on the prior austenite grain size and whether the original precipitates present had tempered. This type of characterization is not technically justified for tempered martensitic steels since properties are controlled by substructure as well as the type and size of precipitates. Recent research has therefore been performed to properly characterize the HAZ regions in 9 wt. % Cr CSEF steels. This research [5,6] involved systematically investigating the microstructural distribution in the HAZ of single pass and multipass welds and performing microstructural simulations under known conditions. The multipass welds capture the accumulated influences from multiple weld thermal cycles.

The advanced characterization techniques used included a Nova 600 Nanolab dual beam (Thermo Fisher Scientific, Hillsboro, OR, USA) focused ion beam field emission gun scanning electron microscope was used to collect the electron backscatter diffraction (EBSD) data from the matrix. Ion-beam-induced secondary electron imaging was used to evaluate the distribution of precipitate particles. EBSD maps were collected using an EDAX Hikari camera (EDAX Inc., Mahwah, NJ, USA), at an accelerating voltage of 20 kV and a nominal beam current of 24 nA.

The output of the characterization made using these advanced techniques is illustrated with reference to the images shown in Figure 7. The observations and subsequent analysis of the results obtained [5,6] demonstrated that a new description for the HAZ regions in martensitic CSEF steels based on the transformation behaviour was required. This description which emphasizes the effect of the welding thermal cycles on the precipitates present is summarized as follows:Completely transformed zone (CTZ)—This region occurs closest to the weld fusion line. Thus, the temperatures range from close to the melting point of the base steel down to about 1020 °C. The combination of time/temperature experienced is sufficient that the original base metal is fully re-austenitized. The application of advanced electron optic techniques reveals that under typical conditions all of the precipitates formed during initial fabrication are dissolved.Partially transformed zone (PTZ)—This region occurs where the peak temperatures range from about 1020 °C to about 950 °C. The combination of time/temperature experienced is such that the original parent metal is only partially re-austenitised. In addition, the application of advanced electron optic techniques reveals that under typical conditions there is insufficient thermal energy to dissolve the pre-existing precipitates.Over-tempered zone (OTZ)—This region occurs where the peak temperatures experienced are below about 950 °C. The combination of time/temperature experienced is such that the original parent metal is not modified when viewed in the optical microscope. However, the application of advanced electron optic techniques reveals the coarsening of secondary precipitates

These descriptions more closely match the original regions described in the documentation from ORNL [19] and are justified by the extensive nature of the characterization techniques used [5,6].

### 3.2. Heat Affected Zone Damage

The heterogeneous microstructures in weld HAZ’s are such that during creep, multiaxial stresses can develop in large cross weld specimens under uniaxial loading. These multiaxial stresses are established because the large samples constrain the deformation until cracking occurs. Metallographic examination of feature test samples has shown that creep damage in the HAZ is a function of the metallurgical risk inherent in the base steel and the local stress state. In the HAZ the welding thermal cycles modify the parent microstructure. The local stress state established during creep is influenced by the weld geometry, orientation and the properties of the specific microstructural zones.

After the creep failure or termination of the creep test, a macro sample was removed from the post-test feature creep test using fine wire, electrostatic discharge machining (EDM). All specimens were removed from the approximate center (the line in the top image in Figure 8) to analyze the most representative distribution of damage well-controlled, feature-type cross-weld creep tests.

When these feature specimens are creep tested to the end of life for low stresses, i.e., long times the fracture path ran through the weld HAZ, Figure 9. Fractures of this nature take place with very little ductility or tearing, and the cross sections shows very little or no reduction in area. Fractures of this type are typical of the damage observed in CSEF welds in service.

Details of the creep damage developed in the specimens have been establish using specialist metallographic techniques. Information regarding the sample preparation and characterization methods have been published [17]. The present paper therefore only summarizes key information used for the laser metallography. However, even a basic appreciation of these factors emphasizes the level of effort invested to obtaining relevant and accurate results. It is apparent that the usefulness of the data is dependent upon careful preparation, observation, recording of data and then analysis. Other researchers are encouraged to comment on specific techniques used in studies documenting the microstructure and creep damage in tempered martensitic steels. The following are the main stages involved with the sample preparation and examination in EPRI research:Following the fine wire electro discharge machining the samples were prepared using metallographic preparation procedures which included initial grinding on 240 to 1200 grit SiC and then subsequent polishing on standard cloths with diamond suspension down to 1 μm. A final extended chemo-mechanical polishing procedure was carried out. This was performed using 0.06 μm colloidal silica suspension. This final polish ensured that all evidence of surface deformation that was introduced in the abrasive stage of preparation was eliminated.A Keyence VK-X105 Confocal Laser Microscope was used for capturing images from the surface of the prepared specimen. The specimen was mounted on a precision stage which facilitated controlled movement and location of a specific region. In the present case the stage used had a maximum travel distance of 100 mm in both the X and Y orientations.For most assessments two pieces of software associated with the Keyence microscope were used for image analysis. These packages were the VK Image Stitching Software and the VK Image Analyzer Software. The VK Image Stitching Software was used to merge the individual images collected into a single compiled image. In general, the images are overlapped by about 12%–15% to ensure that the detailed of the compiled image was accurate. The VK Analyzer Software provided the ability to adjust details of the overall image. Specific examples of the features which could be adjusted include the brightness, contrast, laser intensity and the high dynamic range.

An example of the examination approach is shown in the highlighted region of Figure 10. As with all metallographic techniques there is frequently the need to balance being able to resolve specific items in the specimen and obtaining sufficient images to ensure meaningful observations. Very high magnification can aid in the identification of fine detail, but the small fields involved makes capture of sufficient numbers of features difficult. In the present research a 20× objective was established as appropriate for the analysis of creep cavitation in the HAZ of Grade 91 steel welds. This objective provided a magnification of ~400× on a 15-inch monitor [20].

The magnification is not the only critical variable in the assessment of creep cavitation since the number of pixels in the obtained image can also be altered. The default size for each image collected is 1024 × 768 pixels. The complied macro image can be saved using the full resolution. The number of pixels used for analysis of creep cavitation was 6346 × 3303 pixels. The image size (i.e., the number of pixels) is an important factor in determining the threshold for counting cavities in any material.

The damage developed in cross weld feature testing of a Grade 91 steel which is inherently damage susceptible is illustrated in Figure 11. The creep voids present are shown as dark contrast in Figure 11a. For this feature test, it is apparent that even though high number densities of voids have developed in the weld HAZ there are no cracks present. Thus, for structures which have a uniform susceptibility to void formation, and a similar stress across the section, only very limited periods of stable creep crack growth can take place. The specific region within the HAZ where the highest number of voids was developed is shown in Figure 11b. Based on the detailed analysis performed, it appears that the highest damage levels are found in the region which experienced a thermal cycle in the range 1050 to 920 °C during welding. Whereas, traditional descriptions have termed this as ‘fine grained’ or ‘intercritical’, recent EPRI research has established that this region should more correctly be defined as partially transformed [5,7]. 

## 4. Continuum Damage Mechanics

A creep continuum damage mechanics (CDM) constitutive model for Grade 91 steel developed as part of a multi-year collaborative research project has been used to assess the creep performance of structures and components. As described in previous papers [9,20], EPRI research in the area of CDM has been guided by a systematic, step wise methodology. The following list summarizes the important steps followed during the development and testing of this model, and its ability to describe high temperature behaviour:The foundational step involved identification of relevant creep data for 9% Cr tempered martensitic steels. In addition to identifying results from reputable world-wide organizations, EPRI undertook high temperature testing on well pedigreed Grade 91 and Grade 92 steels. The overall results were used to create a database of creep behaviour that included long term results. The availability of the long-term data with failure lives in excess of 10,000 h limited the need for extrapolation. This increased confidence in using the results to describe in-service behaviour.In addition to initial characterization linking details of composition and fabrication to microstructure, all of the EPRI creep tested specimens were evaluated after testing. Evaluation of these data permitted the rate controlling mechanism to be established. The ability to link stress, stress, temperature and metallurgical variables with creep strain and rupture is critical to ensuring that the formulation of the model is based on good science. Thus, knowledge established regarding these factors was used to underpin the formulation of the equations used for the CDM.The rigorous approach to model development advocated by EPRI ensures not only that the expressions used are justified on a mechanistic basis but also that the approaches are not unnecessarily complicated. Initially, the proposed expressions were tested against overall trends in the results rather than immediately seeking to describe the creep behaviour for a given set of conditions. As described in Section 2, there is definitive evidence that individual heats of steel exhibited specific behaviour. Thus, the formulations and the associated constants were initially tested against the creep behaviour of selected heats of Grade 91 steel tested using simple specimens under constant load conditions.To increase the component relevance of the validation creep behaviour under complicated stress states was assessed. EPRI had undertaken controlled testing using creep specimens with notches. The geometry of the specimens and the notches as well as the applied load control the stresses developed. In the present research the predictions from the CDM were compared against data for more complex stress states.As described in Section 3 of this paper, EPRI research has involved of feature sized specimens. The constraint in these specimens is introduced because the cross-sectional area in the gauge is up 80 times that of a small cylindrical sample. In cross weld tests this constraint introduces complex stress because the microstructural heterogeneity acts as a metallurgical notch.As described in Section 5 of this paper, EPRI research also full-size vessel tests undertaken on components typical of those found in plant. These tests require expert design, fabrication and execution but since all aspects of the tests are well controlled the results provide a unique opportunity for evaluation of predictions.The knowledge developed from root cause analysis of in-service failures offers perhaps the most important opportunity for model validation. These examples of course involve some uncertainty as in an operating power plant specific details of the stress and temperature are not known exactly.

The broad scope of these tasks requires an integrated and sustained approach implemented and reviewed over several years. The basis for this research has been described in an EPRI published report [9]. The overall plan required that there would be key points over the course of the model development and validation where the details of the work were reviewed and further evaluated. At each of these review points, important knowledge gaps or needs, such as the requirement for additional metallurgical data, were identified and the necessary testing and analysis undertaken.

Using the logical approach outlined above the expression for the CDM was formulated recognizing that the creep deformation and fracture behavior was different at high stresses and at low stresses. The relationship used to describe the stress dependence of the minimum creep rate involved summing two power laws. This overall relationship is shown in Equation (1). This considers the two different high temperature damage mechanisms, namely strain softening and cavitation.
(1)$${\dot{\varepsilon }}_{min}=A_{HT}\sigma^{n_{H}}+A_{MT}\sigma^{n_{m}}$$
where
(2)$$A_{HT}=A_{H}exp\left(-Q_{H}/RT\right)$$
and
(3)$$A_{MT}=A_{M}exp\left(-Q_{M}/RT\right)$$

In these equations, ${\dot{\varepsilon }}_{min}$ is the minimum creep rate, *R* is the ideal gas constant, *Q* represents the Activation Energy and *T* is the temperature in Kelvin.

The damage and strain softening state variable are incorporated into the strain rate equation as shown in Equation (4). Damage as a result of strain softening was defined by the state variable G, which is 0 in the initial condition and when ductile failure occurs. The softening rate is proportional to the strain rate, see Equation (5). Cavitation damage was described associated with the formation and growth of creep cavities. Following the work of Kachanov [21], cavitation was represented by the state variable, ω, and this also varies from 0 to 1. The classical damage rate equation with (1 − ω) is used to describe the relationship between the rate of cavitation and the value of the maximum principal stress, see Equation (6).
(4)$${\dot{\varepsilon }}_{ij}^{c}=\frac{3s_{ij}}{2\sigma_{e}}\left(A_{HT}\sigma_{e}{}^{n_{H}}+A_{MT}{\left(\frac{\sigma_{e}}{\left(1-\omega \right)\left(1-G\right)}\right)}^{n_{M}}\right)$$
(5)$$\dot{G}=k{\dot{\varepsilon }}_{e}^{c}$$
(6)$$\dot{\omega }=\frac{A_{DT}\sigma_{I}^{x}}{{\left(1-\omega \right)}^{\varphi }}$$

In these equations $\dot{\omega }$ is the rate of damage due to cavitation,$\sigma_{I}$ is the maximum principal stress, $\dot{G}$ is the strain softening rate, ${\dot{\varepsilon }}_{e}^{c}$ is the equivalent creep strain rate, ${\dot{\varepsilon }}_{ij}^{c}$ is the creep strain tensor, $s_{ij}$ is the deviatoric stress tensor, and $\sigma_{e}$ is the von Mises equivalent stress.

Extensive analysis of the Grade 91 heat affected zone microstructure by EPRI has identified distinct microstructural regions in the HAZ. The first, termed the completely transformed zone (CTZ), is located adjacent to the weld fusion line. This zone has historically been referred to as the coarse-grained heat-affected zone is exposed to the highest temperatures during the welding process and the microstructure which develops in the CTZ is similar to the microstructure observed in the base metal. For this reason, in CDM analyses of cross-weld behavior, material properties in this zone have been assumed to be the same as those of the base metal. The second zone is termed the partially transformed zone (PTZ). The temperatures reached in this region are less than those achieved in the completely transformed zone and a desired dislocation and precipitate substructure is not present. The PTZ is associated with lower strength and greater susceptibility to creep cavitation than in Grade 91 base metal or the CTZ.

The structure of the constitutive model for the partially transformed zone is the same as that of the Grade 91 base metal model. In order to represent the difference in strength between the partially transformed zone and the base metal, a strength factor *F* was introduced, where 0 < *F* ≤ 1. When *F* = 1, base metal strength is achieved; when *F* < 1, the creep strength is less than that of base metal:(7)$$\dot{\varepsilon }=A_{HT}{\left(\frac{\sigma }{F}\right)}^{n_{H}}+A_{MT}{\left(\frac{\sigma }{F\left(1-G\right)\left(1-\omega \right)}\right)}^{n_{M}}$$

The strength difference observed between the partially transformed zone and the base metal is described using the factor, *F*. Since *F* is a function of temperature an Arrhenius temperature dependence was incorporated as follows:(8)$$F=1-F_{0}exp\left(-Q_{F}/RT\right)$$

The stress redistribution leads to enhancement of the maximum principal stress in the PTZ. This was considered when determining material parameters for the damage mechanism.

Determining material property coefficients for the PTZ model requires data from a minimum of three test types (each performed over a range of temperatures and stresses): uniaxial smooth bar tests of the base metal to define base metal creep strength; uniaxial smooth bar tests of simulated PTZ material to define the creep strength of the PTZ; and a test imposing triaxial constraint on the PTZ or simulated PTZ material to determine the cavitation damage response. At present, cross-weld tests have been used to define the damage response of the PTZ. To ensure the coherence and relevance of the developed data, all the proceeding tests should be performed on material from the same heat.

Testing according to the above methodology was performed by Hongo et al. [22], representing a coherent and relevant dataset that could be used to determine material parameters for the partially transformed zone (note that the partially transformed zone is referred to as the fine-grained HAZ in [22]). A comparison of the measured and predicted time to rupture for uniaxial creep rupture tests of Grade 91 steel base metal, simulated HAZ (PTZ), and cross-welds is shown in Figure 12. A key item to note is that the model predicts failure by the development of damage in the HAZ for cross-welds tested at low stresses, but failure by creep rupture of the base metal at high stresses. The behavior is thus directly in accordance with the trends noted in tests reported elsewhere [22].

As an example of the application of CDM to the behaviour of a component, the physically-based creep continuum damage constitutive model was applied to the serviceability assessment of a 90° large bore, welded branch connection [9]. This connection had developed a steam leak which occurred as the result of in-service cracking. The connection was removed from the component and selected samples were metallographically prepared and examined in detail, Figure 13. The model based predicted time to crack initiation (when), location of crack initiation (where), and direction of subsequent crack growth (how) were shown to agree with the observed trends in the reported failure. The agreement between observation and analysis underpins confidence that cracking initiated at the surface. This indicates that nondestructive examination techniques such as surface metallurgical replication and magnetic particle testing can be used to detect damage. However, some caution is required since the operating time between initiation and through-wall cracking can be short. For the exemplar component a leak-break-before break assessment was performed to provide an estimate of damage tolerance, i.e., an estimate of how long the crack would take from initiation to through wall growth. This further analysis demonstrated that the crack growth rate was relatively slow and there was a very low risk of the connection to fracture catastrophically. The expectation for a leak type failure was based on the fact that crack extension of greater than 90° around the branch pipe would be necessary for catastrophic rupture of the connection to occur. The prediction of leak not break is consistent with experience of cracks in similar components. Damage for the typical component geometries is found by steam leaks, not catastrophic rupture.

## 5. Advanced Weld Repair Technologies

EPRI has a long-established reputation in the development and validation of weld repair methodologies. Research supporting fossil-fired asset management has developed viable weld repair techniques for mainstay power generation CrMo steels which have been widely used for coal-fired boiler or combined cycle heat recovery steam generator (HRSG) components. A series of reports provide research results and document weld repair procedures specific to CrMo steel Grades 11, 12 and 22 [23]. It is clear the knowledge base published provides an important strand for cost-effective life management, i.e., informed run/repair/replace decision-making. The research performed through the EPRI studies, in part, led to the acceptance of Welding Method 4 and Welding Method 5 in the NBIC Part 3 Repairs and Alterations. These weld repair methods are often cited as ‘temperbead methods’. Although the primary consideration for CrMo and CrMoV repair is the avoidance of reheat cracking through the concept of ‘grain refinement’ rather than simply tempering the microstructure.

EPRI-coordinated, industry-sponsored research projects in CSEF steels began in 2007 with a major effort to establish sound life management approaches for Grade 91 steel components. The learnings and findings, realized with direct input and perspective from over 40 stakeholders representing the entire electricity supply chain, underpin component specific repair methodologies for the CSEF steel Grade 91 in use today. Key evidence for these ‘alternative’ repair methods are summarized below.

### 5.1. Development of Alternative Repair Methodologies

Historically, it has often been the case that weld repairs performed after periods of in-service operation were based on the procedures mandated for new construction. Clearly since the equipment is not new it is important to consider whether this approach is reasonable. A major EPRI initiative investigating and quantifying the factors affecting weld repair of CSEF steels started in 2011. After this initial thrust, a series of research programs has been undertaken to develop well-engineered weld repair procedures, initially for Grade 91 steel components [10]. The variables studied in these studies are summarized in Table 2. It is important to emphasize that this research resulted in acceptance of weld repair procedures which could be applied without the need for PWHT. The development of weld repair methodologies has been largely proactive, so that accepted procedures were established before widespread failures were reported. Today, the research continues to evolve the repair methods for dissimilar metal welds to Grade 91 steel and for the next generation of 9 to 12%Cr CSEF steels such as Grade 92.

The power generation industry is now recording operating times in boilers and piping systems using Grade 91 steel which exceed 100,000 h. The likelihood for damage increases with increasing operating time, so that as time in service increases so does the need for more regular and widespread repairs. More than two dozen EPRI reports underpin the recommended repair practices now recognized by the NBIC Part 3 Repairs and Alterations as Welding Method 6 (first published in 2015 edition) and Welding Supplement 8 (first published in 2017 edition). These methods include options without a mandatory post weld heat treatment (PWHT). Examples of publicly available information can be found EPRI documents for example [10,24,25].

The design and implementation of well-engineered weld repairs in Grade 91 steel components required a number of considerations be taken into account to provide a best practice approach. It was apparent that there is no one-size-fits-all procedure for all alternative weld repairs. The critical decisions and recommendations that were reviewed during the code approval process for Welding Method 6 and Welding Supplement 8 have been explained in a key document. Some of the important questions raised and answered during the code approval process have been documented [26]. These summaries concerned the following issues on performance:Responsibilities of authorized inspectors and jurisdictions;Welding qualifications and monitoring repair quality;Documenting repair information and weld repair performance;Key definitions such as “well-engineered repair” and “impractical”;Filler metal selection;Defect removal and excavation practices and partial or localized excavations;Applicability to creep strength enhanced ferritic steels other than Grade 91;Susceptibility to stress corrosion cracking;Limited access repairs;Damage and root cause analyses;Using a minimum post weld heat treatment of 1250 °F (675 °C).

The knowledge base supporting these repair methods will be updated periodically as needed to address questions specific to new or emerging issues associated with alternative weld repairs in Grade 91 steel.

### 5.2. Validation of Alternative Repair Methodologies

Although significant information was produced to validate the alternative repair methods during the first phases of research on-going studies include refinement of performance expectations for the weld repairs. These further approaches for quantification of behavior include the use of state-of-the-art testing and evaluation methods to provide a comprehensive basis for informed decision making. These independent approaches include:feature-type cross-weld creep testing including assessment of the creep behaviour of dissimilar metal welds (Figure 14);full-size vessel tests undertaken on components typical of those found in plant (Figure 15), andcontinuum damage mechanics and constitutive modeling of the performance of repairs in the structural components (Figure 16).

A typical feature test cross weld creep specimen in shown in Figure 14A. This specific example shows a weld which simulates the manufacture of repair in a thick section joint between Grade 91 and Grade 22 steels. The width of the repair is purposely designed to extend beyond the HAZ of the original weld profile. This extended fill creates a step in the repair profile, and this has been shown to increase the damage tolerance of the joint. As shown in Figure 14B, creep damage which develops in the HAZ of the repair cannot easily propagate because the base metal exhibits greater resistance to creep damage than the HAZ. This is a very important benefit because in the majority of cases weld repair is required after removal of damage, i.e., the location needing to be repaired is known to be susceptible to forming damage. In the current case, the weld HAZ has developed stable micro and even macro defects which should be readily detected using recommended nondestructive testing methods.

An example of one of the EPRI designed vessel creep tests is given in Figure 15. This vessel was manufactured from a cylindrical section of Grade 91 steel taken from a superheater outlet header. The vessel contains several features representative of the geometric complications found in power and petro-chemical plant. The features included in the vessel were a flat end cap and an end plug, a longitudinal seam weld and stub tube to header welds. In the original pressure boundary circumferential weld, a partial weld repair was made using one the newly approved repair approaches. The repaired region is shown in the schematic section of Figure 15. As previously the excavation and subsequent fill extend beyond the HAZ of the original weld. This vessel has been on test at 625 °C and the results will be reported in detail in due course.

The vessel type creep tests offer very valuable direct evidence of high temperature performance both in terms of the time to cracking, the locations of first cracking and cracks growth leading to failure. As these tests are undertaken in purpose-built facilities it is possible to run tests to fracture as all risks are minimized. It is, however, not possible to perform these specialist experiments on all combinations of pressure, temperature and vessel design. Thus, in addition to extracting maximum value from the direct experimental observations further benefits are derived if the actual results from the vessels are assessed using the CDM analysis methods described earlier. The EPRI recommended approach has been to seek to benchmark this type of analytical assessment using results from in-service Case studies and vessel data. Figure 16A shows the results of stresses developed in the vessel as a consequence of internal pressure. Details of damage analysis for the HAZ of pressure boundary welds are presented in Figure 16B.

A further important method for evaluation of performance involves collating and analysis of information regarding the behaviour of actual repairs in installed components. Development of this database permits analysis of the performance of the overall performance of repairs and more detailed assessment by component. In the case of Grade 91 steel, EPRI has documented thousands of repairs over the last five years in multiple counties which were fabricated in accordance with the NBIC and EPRI reports. Tracking these cases is vital for the industry to provide best practices should issues arise, and such that the continued success of specific approaches is supported by statistical relevant information.

Future work is necessitated to evaluate repair options for other CSEF steels allowed by Design Codes. These include Grade 92, and a new generation of emerging CSEF steels such as VM12SHC (ASME Code Case 2781), Grade 93 (ASME Code Case 2839) and THOR 115 (ASME Code Case 2890). Furthermore, common place failures in traditional and A-ASS materials in tube to attachment welds is necessitate the development of novel test techniques to screen material performance and provide insightful guidance for remediation using weld repair.

## 6. Discussion

### 6.1. Creep and Fracture of Welds

Assessment of the creep performance of welds is frequently studied using cross weld testing. Critical issues in setting up these programmes include selecting test conditions and specimen geometries that result in damage mechanisms which are relevant to long term service. It is now widely accepted that the results of creep tests at relatively high stress and temperature are not typical of long-term damage in component welds. Thus, information from tests under non-representative conditions should not be used as a guide to in-service behaviour. EPRI has established a testing protocol which involves performing cross weld creep tests on feature test type specimens. These sample examined in this study has a cross sectional area in the gauge of ~965 mm^2^ (1.5 inch^2^). This is about 30× greater than typical cross-sectional areas for standard round bar creep tests machined to a diameter of 6.35 mm (0.252 inches).

The characterization of damage in these cross weld tested samples typically involves [17,27] visual examination followed by optical metallography. The microscopy was carried out on sections which were prepared using careful polish/etch techniques to reveal damage present on the central plane of each specimen. This approach has been shown to be necessary since the tendency to form service relevant HAZ creep damage is related to hydrostatic stresses and is thus minimized at the specimen surfaces.

Experience has shown that as a rule the welds present in the pressure boundary of CSEF steel components are susceptible to creep damage and fracture before base metal locations. Observations of this type have therefore focused research to evaluate the factors affect weldment performance. It is historically the case that research of this type involves undertaking laboratory creep tests of specimens cut from the selected welds. In most cases decisions regarding the geometry of the samples are made considering the size of the welded section available and the load capacity of the available creep test machines. In most cases then the specimen geometry chosen is that of a cylindrical section with a diameter in the range 4 to 10 mm. EPRI has established a testing protocol which involves performing cross weld creep tests on feature test type specimens. The types of specimen involved in EPRI research assessing the creep behaviour of welds are shown in Figure 17. These samples typically have a cross sectional area in the gauge of ~965 mm^2^ (1.5 inch^2^). This is about 30× greater than typical cross-sectional areas for standard round bar creep tests machined to a diameter of 6.35 mm (0.252 inches). Figure 17 compares the feature type specimens with a small cylindrical sample of the type used in other testing.

In addition to selection of a meaningful specimen geometry it is also important to consider the test conditions since all of these factors will influence whether damage mechanisms in the laboratory tests are relevant to long term service. Experience has shown that the results of cross weld creep tests which are performed at relatively high stress and temperature are not typical of long-term damage in component welds. Thus, information from tests under non-representative conditions should not be used as a guide to in-service behaviour. It is critical then that the details of the fracture location and relationship of the damage present with the constituent microstructure of cross weld tests is established using accurate posttest metallographic preparation and examination. This type of characterization provides important knowledge as to how to use cross weld creep test results.

Details of the post-test characterization methods recommended by EPRI regarding the sample preparation and characterization methods have been published previously [17]. It is clear that to accurately document the creep damage developed in the specimens requires the use of specialist metallographic techniques. The information summarized in the present paper provides an important appreciation of the level of effort invested to obtaining relevant and accurate results using laser metallography. It should be emphasized that the usefulness of the data is dependent upon careful preparation, observation, recording of results and then analysis. This process must start by ensuring that the sectioning is performed to reveal damage present on the central plane of each specimen. This is necessary since the tendency to form service relevant HAZ creep damage is related to hydrostatic stresses and is these hydrostatic stresses are minimized at the specimen surfaces. In addition to the need for care during sectioning and specimen preparation, it is important when documenting creep cavity size and shape that no etching of the specimen is undertaken. It is clear that etching will modify both the size and character of cavities. In general, etching will increase the size of cavities present and will tend to ‘round’ the cavity shape. While the use of laser microscopy provides a reasonable record of the number density of creep voids it is clear that the most accurate observations of creep void shape are obtained using ion beam techniques [17,27]. In this advanced research, the cavities are shown to be angular in shape and usually associated with the inclusions or other hard particles present.

In addition to care in preparation, examination and recording of the creep cavities present effort should also be invested in careful selection of both magnification and image resolution. Earlier research [17] has shown that for images recorded from the same specimen using the same magnification, the cavitation densities shown were markedly different as a consequence of image resolution. Thus, for the lower resolution used, the theoretical minimum cavity diameter that was counted was 2.04 µm. In contrast, using an improved image resolution, improved the minimum cavity diameter that was counted to 0.70 µm. Because the digital processing of images using light microscopes is becoming increasingly common, it is critical that that appropriate settings and procedures are selected. In the case of the equipment used in EPRI studies, there are options for utilizing “super fine” image capturing that would further increase the resolution of the selected objective to 0.34 µm. The difference in cavity size recorded between the low resolution and “super fine” resolution represents a range of almost an order of magnitude.

The parent metal condition has a direct effect on the cross-weld creep performance in Grade 91 steel. This is most often put into the context of a deformation-related mechanism (i.e., strength), but as presented in this review paper there is a clear influence of the damage resistance (i.e., ductility) on cross-weld creep performance. Thus, the results from EPRI research demand that designers and alloy developers place an equal emphasis on parent metal creep strength and ductility. Inherent base metal performance is a key aspect linked to the susceptibility to damage in the HAZ. Detailed testing supported by comprehensive metallurgical characterization has established the factors which influence cross-weld creep performance. For a set of Grade 91 materials which exhibited the same strength, an increase in the ductility from 15% to 83% ROA resulted in increased the cross-weld creep life by 5× at a test condition of 625 °C (1157 °F) and 60 MPa (8.7 ksi). Ongoing assessment is working to more definitively link risk factors in the microstructure which are responsible for damage (i.e., inclusions and/or other cavitation-susceptible features in the material). However, it is apparent that although weld performance maybe life limiting for in-service components a significant contribution effecting the life and manner of creep fracture can be directly attributed to the base steel. As shown in Figure 18, the creep performance of cross weld tests and of simulated HAZ microstructures are very similar under low stress long–time conditions.

The effect of filler metal strength on weldment performance has also been included in EPRI research. Cross weld creep tests on samples made using a weld filler metal with an over-matching strength to the base metal did not show any trend on life. For example, there was no significant influence on life for creep tests performed at an applied stress of 60 MPa (8.7 ksi). In the tests at a higher applied stress of 80 MPa (11.6 ksi), there was some evidence of reduced creep life as filler metal strength increased but the results of all tests were still above the lower bound of acceptable performance, i.e., the test duration was greater than the mean-20% bound. In contrast, cross-weld testing of samples made with under-matching filler material consistently failed at the mean-20% bound, Figure 18. It appears that the fact that welds with lower creep strength filler show creep failure in the HAZ can be explained by the fact that the B8 type filler material used was close to the strength of the PTZ region of the HAZ. It was also of potential practical benefit to note that the cross-weld tests for the under-matching filler exhibited the greatest damage tolerance of all the weld metals studied. Thus, in these tests there appeared to be evidence that macro-cracking started at an earlier life fraction than in tests with the higher strength filler metals. It is clear that a relatively slow rate of crack propagation increases the opportunity for defect detection by non-destructive testing during service. It is apparent therefore that designers should consider the geometry of the weld and the specific properties of the filler metal on component creep performance. Ensuring damage tolerance is a key benefit to component design for long-term operation under high temperature creep conditions.

### 6.2. Management of Assets Made from CSEF Steels

The life management approach at the fleet-, plant-, system- or component-level is conventionally divided into a staged approach [24]. The Level 0, I, II or III methodology incorporates important concepts such as risk-ranking or risk-based inspection (RBI) and fitness for service (FFS). The level approach is summarized in the following information:Level 0. Macro screening and risk-ranking of utility fleet and/or of the systems in the plant.Level I. Conservative and refined risk-ranking of susceptible plants and/or components in the system.Level II. Assessment of plant-specific information to provide information on ‘where’ and ‘when’ to look for damage in the plant, system or location in the system.Level III. Detailed assessment of plant-specific information and consideration of unique material properties including the potential disposition of defects in the system.

Fundamentally, this level of detail helps to refine prior recommendations and assess ‘how’ the component will fail, e.g., leak or break. The end user must be involved in performing a meaningful risk assessment, particularly to ensure that the consequences of fracture are properly established and factored into the risk calculation.

As a minimum EPRI recommends that defining the consequence of the event should consider the following factors in relation to component failure:Safety. Proximity of the location to a high traffic area or control room, etc.;Collateral damage. Such as the proximity of the location to a critical component or support equipment that may be damaged in a leak or failure;Financial impact. This can include lost generation, replacement generation or other considerations that are difficult, and in many cases impossible, for a vendor or independent assessor to define.

Risk-ranking can be performed to a progressively more detailed level of probabilistic and consequential assessment which typically requires more certain engineering factors (e.g., stress/temperature/metallurgy/dimensional measurements). Basic concepts which should be considered in the risk-ranking of 9%Cr steel components are covered in more detail in [10,12,28,29]. The general approach to risk-ranking is defined in ASME Post Construction Code 3 (PCC-3) with a more thorough approach defined in EN 16991:2018 [29]. Initially, effective asset management should target components which have a high expectancy to fail early-in-life. This decision-making is predicated on identification of the factors which contribute to reduced performance (e.g., design, fabrication, operation and metallurgy). In many cases, a detailed understanding of the historical issues identified through available service experience reports, such as in [13,14], are of immense value to the engineer to provide needed perspective and begin to target the highest risk locations

## 7. Concluding Comments

Components in power generating plants must operate safely under complex conditions, including a high temperature, high pressure and severe environments. New or post construction codes provide a basic set of rules to ensure that a minimum expected performance will be achieved. Fabrication of power generation components necessitates joining sections to establish a suitable pressure boundary through conventional fusion welding processes. Throughout the course of a component’s life, damage at the most susceptible, high-risk locations will necessitate run/repair/replace decision-making as part of an integrated life management philosophy.

The present review emphasizes that the high temperature performance of tempered martensitic steels is complex. However, by appropriate planning and introducing good practice procedures, significant benefit to ‘through life’ costs can be achieved. These savings can be realized without having to compromise equipment safety and reliability.

## Figures and Tables

**Figure 1 materials-12-02257-f001:**
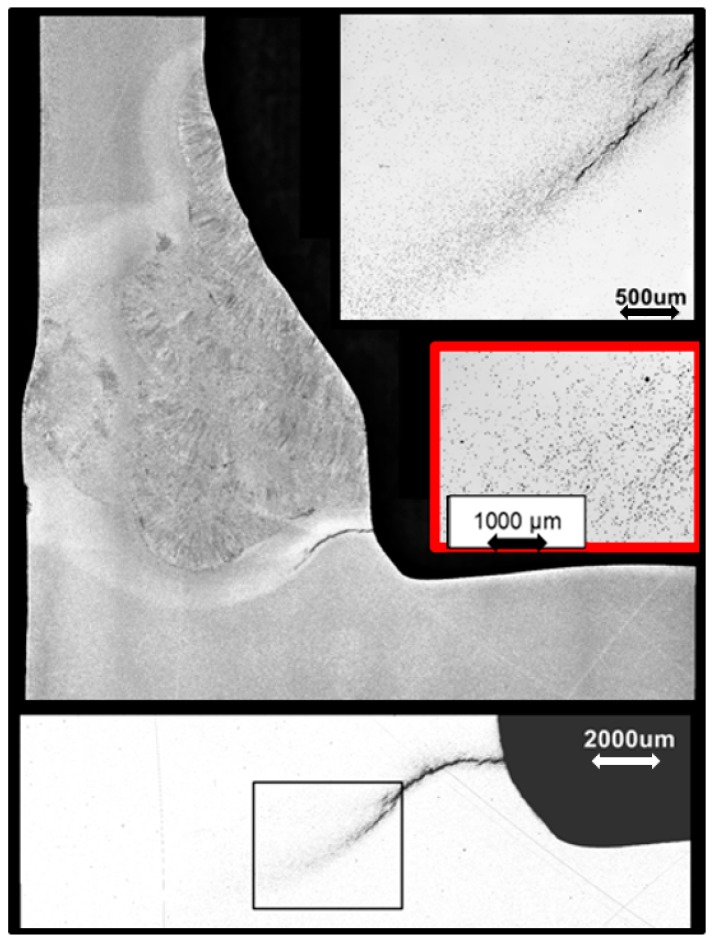
Micrographs showing detail of cracking at a stub tube to header weld. Cracking found at this location at an earlier inspection had been removed by local grinding. It is noteworthy that the re-cracking event did not take place at the base of the excavation. Instead, the creep damage is focused on the metallurgically susceptible structures in the weld HAZ.

**Figure 2 materials-12-02257-f002:**
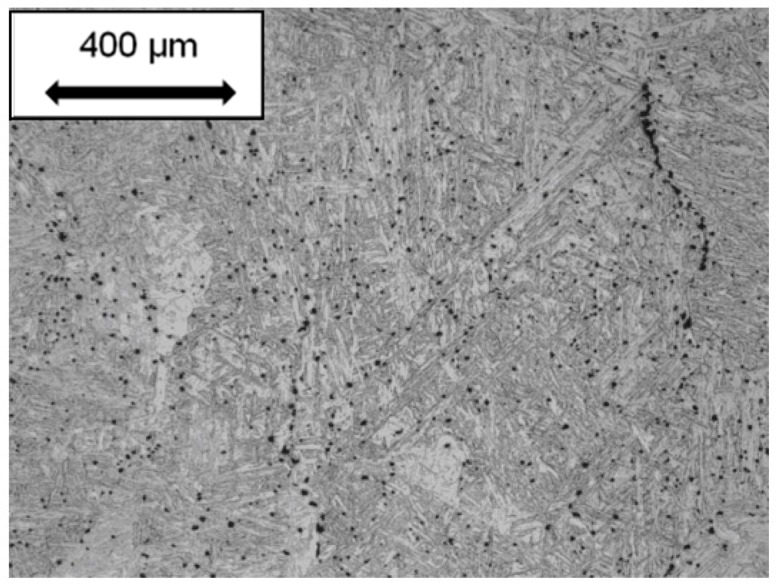
Micrograph showing detail of creep cavities formed in the base metal of a Grade 92 CSEF steel. These creep voids nucleate at a size below 1 μm and grow during component life. The sample shown here is close to final fracture, yet no macro cracks had formed.

**Figure 3 materials-12-02257-f003:**
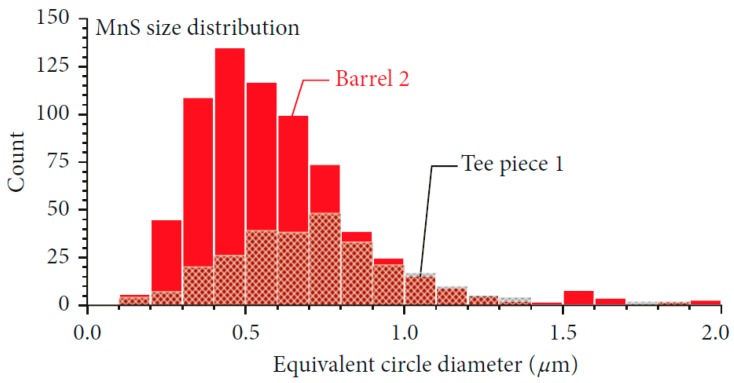
Number density of inclusions for two different but ASME code acceptable Grade 91 steels. A higher density of MnS was measured in the creep damage susceptible steel which contained 0.009 wt.% S (labelled barrel 2) as compared to the cavity resistant steel which contained 0.002 wt.% S (labelled tee piece 1).

**Figure 4 materials-12-02257-f004:**
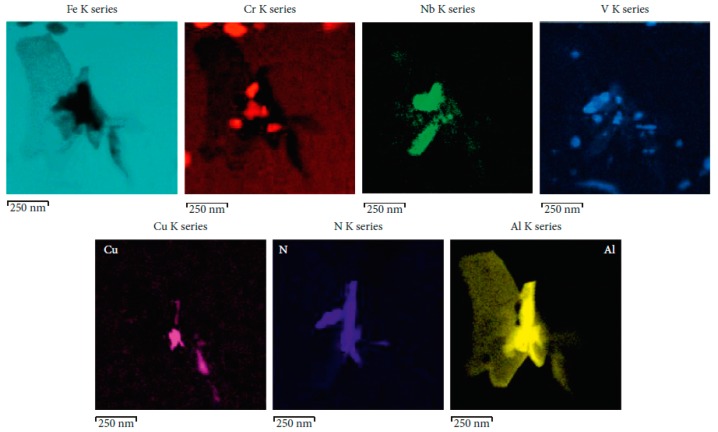
Detailed micrograph showing inclusions associated with an individual creep cavity as well as compositional maps showing the distribution of the elements Cr, Nb, V, Cu, N, and Al associated with this location.

**Figure 5 materials-12-02257-f005:**
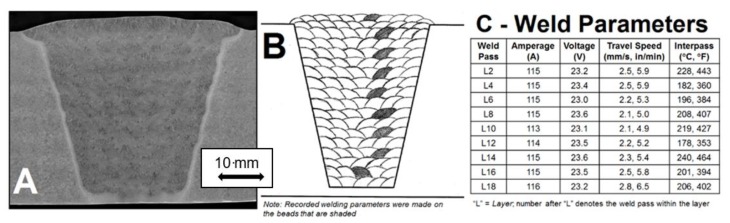
(**A**) Macro Sample of the As-fabricated Weldment in the Post Weld Heat Treated Condition. (675 °C, 1250 °F for 2 h) as shown the pipe thickness is 50 mm; (**B**) Fill Sequence used to complete the Weldment. Note that the darkened fill passes constitute a fill pass that was monitored for voltage, amperage, travel speed and interpass; (**C**) Details for the Monitored Fill Passes.

**Figure 6 materials-12-02257-f006:**
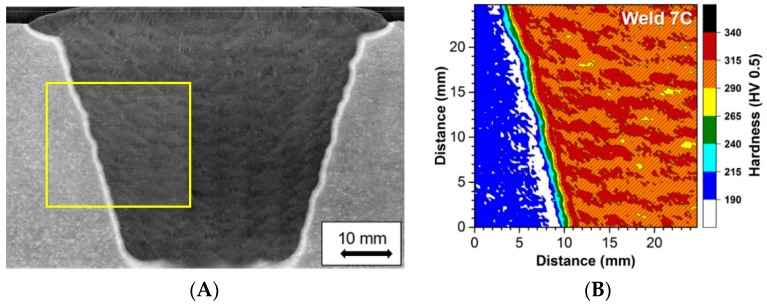
(**A**) Macro Sample of the As-fabricated Weldment in the Post Weld Heat Treated Condition (675 °C, 1250 °F for 2 h) with the yellow box showing the region examined by hardness mapping; (**B**) The hardness map produced with a scale showing hardness ranges for each colour.

**Figure 7 materials-12-02257-f007:**
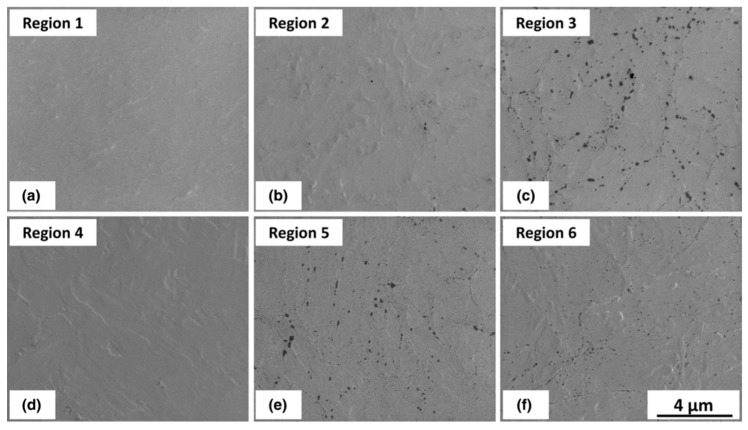
Ion-beam-induced secondary electron micrographs (**a**–**f**) showing the secondary precipitate particles in the HAZ.

**Figure 8 materials-12-02257-f008:**
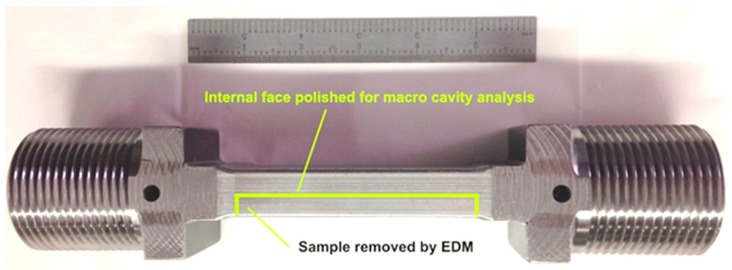
Example of Feature Creep Tests Used in the Evaluation of Damage in the Heat Affected Zone of Grade 91 Steel. Note: the feature test sample contains the entirely of the weld, the HAZ on both sides of the weld and sufficient base metal to promote stress states in the weldment.

**Figure 9 materials-12-02257-f009:**
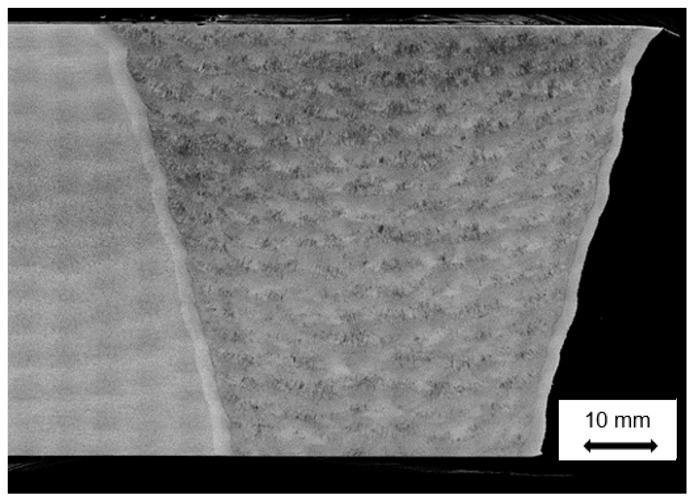
Macro Image from a feature sample of Grade 91 Weldment which was creep tested to failure.

**Figure 10 materials-12-02257-f010:**
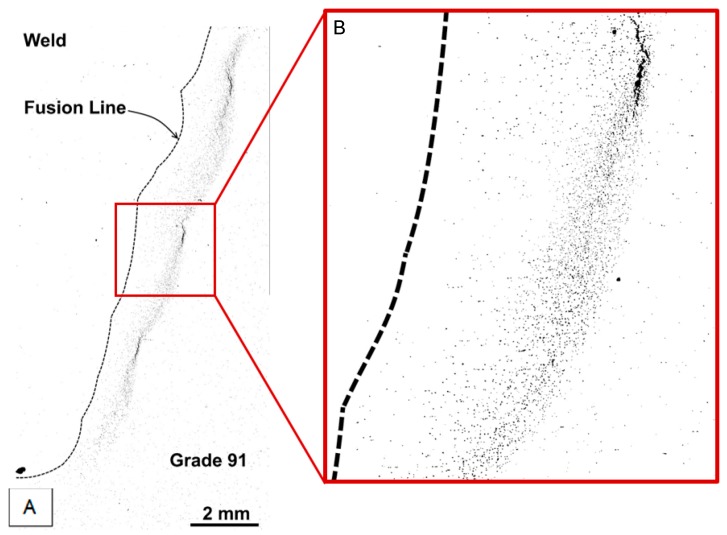
(**A**) Macro Image from the Examined Grade 91 Weldment and Highlighted Region of Interest along the fusion line and HAZ of the Weld; (**B**) Magnified View of Creep Damage in the HAZ of the Weldment.

**Figure 11 materials-12-02257-f011:**
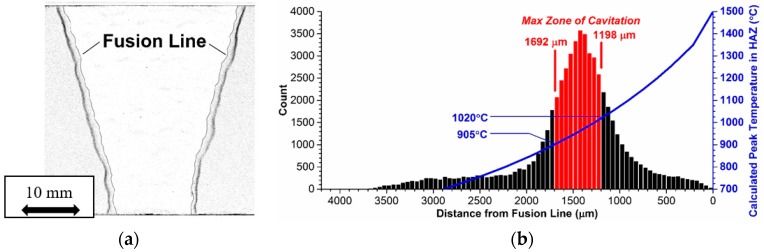
Macrograph showing a Grade 92 steel cross weld feature test sample after long term testing, note using the EPRI classification system the steel parent is considered as ‘susceptible’ to void formation (**a**) and, histogram showing the number of voids present as a function of the distance from the weld fusion line (**b**). The basic EPRI procedures used for sample preparation, data capture and analysis have been detailed previously [17].

**Figure 12 materials-12-02257-f012:**
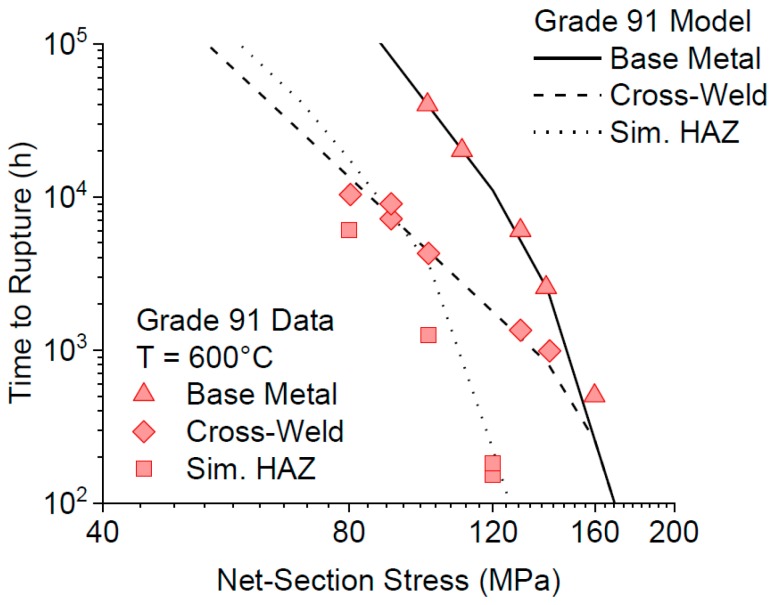
Comparison of measured and model predicted time to rupture for Grade 91 base metal, simulated HAZ, and cross-welds all tested at 600 °C and under uniaxial loading. The data shown in this Figure were published previously [22], the lines shown are as estimated from the CDM analysis.

**Figure 13 materials-12-02257-f013:**
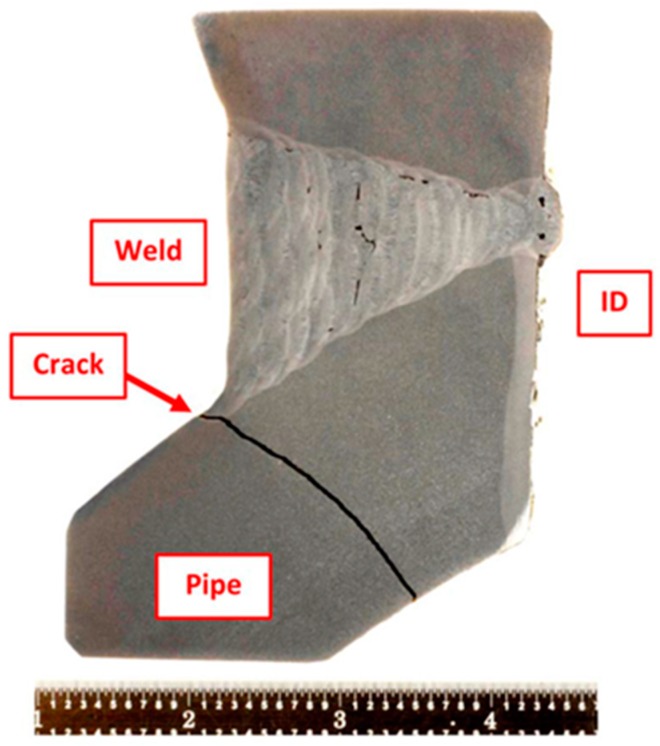
Metallographic section showing in-service cracking at a welded tee piece connection. The scale shown in graduated in inches.

**Figure 14 materials-12-02257-f014:**
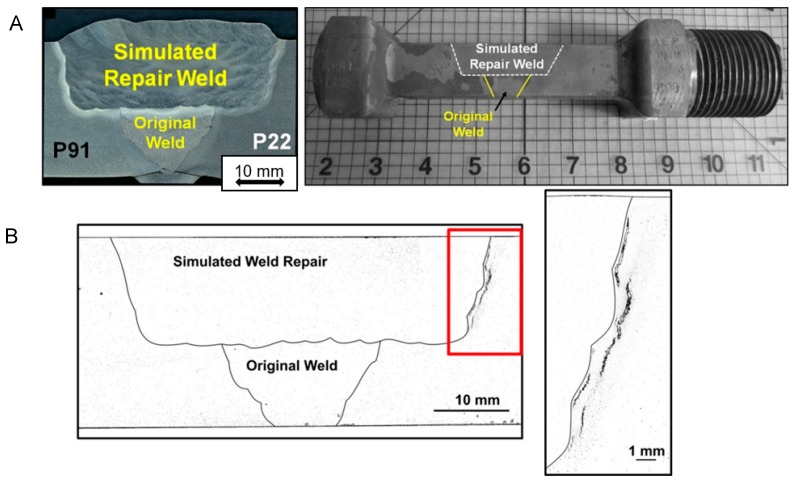
Example of a feature test cross weld specimens used by EPRI, (**A**) containing a simulated weld repair and (**B**) showing HAZ creep damage in a dissimilar metal repair weld between Grade 91 and Grade 22 Steels.

**Figure 15 materials-12-02257-f015:**
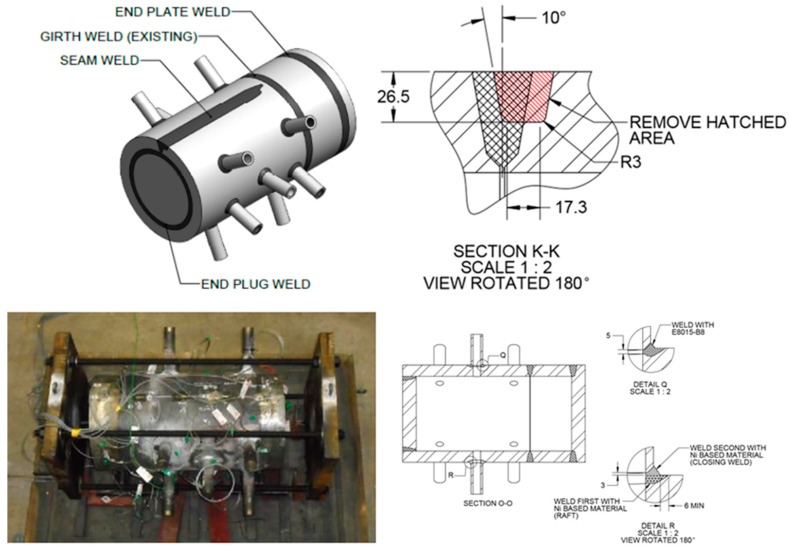
Example of a full-size component vessel test containing innovative weld repairs in Grade 91 Steel performed without post weld heat treatment.

**Figure 16 materials-12-02257-f016:**
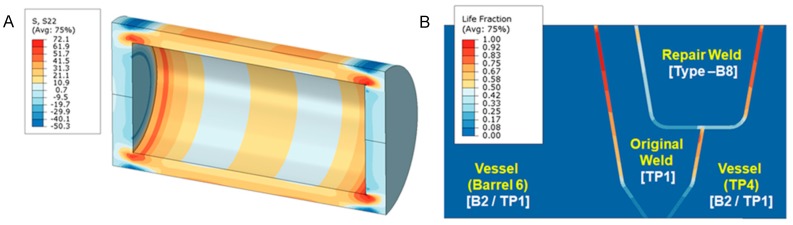
Evaluation of vessel test behavior using finite element analysis coupled with physically-informed constitutive models for creep damage initiation and crack growth, (**A**) showing local stress concentrations at the end caps and (**B**) showing damage estimates for the weldments.

**Figure 17 materials-12-02257-f017:**
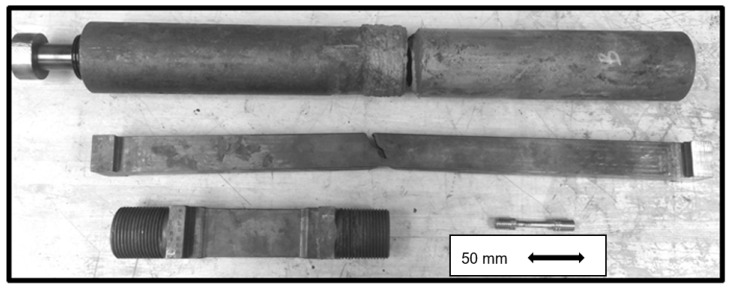
Examples of the feature test cross weld specimens used by EPRI to evaluate the factors affecting the creep behaviour of pressure boundary joints in tempered martensitic steels.

**Figure 18 materials-12-02257-f018:**
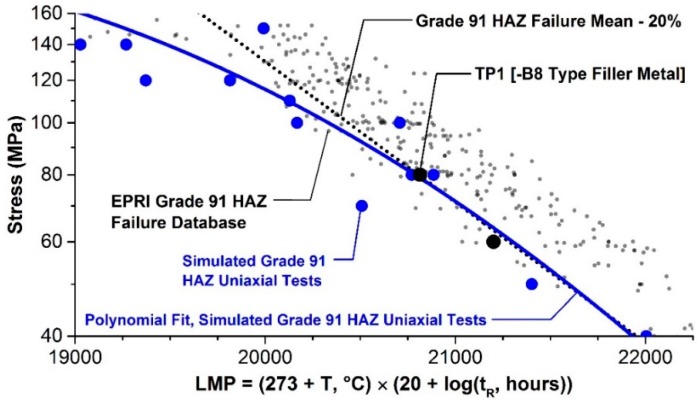
Relationship showing the creep performance of the HAZ defines the properties of cross weld specimens in long times.

**Table 1 materials-12-02257-t001:** Composition for Controlled Quality Grade 91 steel specified in ASME Code Case 2864 [16].

**C**	**Mn**	**Cr**	**Ni**	**Mo**	**V**	**Nb**	**N**	**B**	**W**
0.08–0.12	0.30–0.50	8.0–9.5	0.20 max	0.85–1.05	0.18–0.25	0.06–0.10	0.035–0.070	0.001 max	0.010 max
**Si**	**Al**	**Cu**	**S**	**P**	**As**	**Sn**	**Sb**	**Pb**	–
0.20–0.40	0.020 max	0.10 max	0.005 max	0.020 max	0.010 max	0.010 max	0.003 max	0.001 max	–

**Table 2 materials-12-02257-t002:** Variables Studied in the Examination of Well-Engineered Repair Methods in Grade 91 Steels.

Excavation	Base Metal	Weld	Weld Metal	Process
Minor (50% depth)	Tee Piece 2 to Barrel 3 (Groove in Barrel 3)	1C	E9015-B91	675 °C PWHT
		2C	E8015-B8	Controlled Fill
		3C	EPRI P87	Controlled Fill
Partial (50% depth)	Barrel 1 to Tee Piece 1	4C	E9015-B91	675 °C PWHT
		5C	E8015-B8	Controlled Fill
		6C	EPRI P87	Controlled Fill
Full Repair (~95% depth)	Tee Piece 1	7C	E9015-B91	675 °C PWHT
		8C	E8015-B8	Controlled Fill
		9C	EPRI P87	Controlled Fill
Full Repair (100% depth)	Barrel 1 to Barrel 2	10C	E9015-B91	675 °C PWHT
		11C	E8015-B8	Controlled Fill
		12C	EPRI P87	Controlled Fill
Refill+ Repair (50% depth)	Barrel 1 to Barrel 2 (Groove in Barrel 2)	13C	E8015-B8	Controlled Fill
		14C	EPRI P87	Controlled Fill
Partial (25% depth)	Tee Piece 2 to Barrel 3	15C	E8015-B8	Controlled Fill
		16C	EPRI P87	Controlled Fill

Note: Welds made using Controlled Fill were NOT heat treated prior to testing.

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
