# Peer review of "Manufacture and Performance of Welds in Creep Strength Enhanced Ferritic Steels"

_materials, 2019, doi:10.3390/ma12142257_

Round 1

Reviewer 1 Report

The authors present an interesting synthesis of the research carried out at EPRI to investigate the creep behavior of ferritic steels welded joints. The welded joints are deeply studied both from microstructural and mechanical points of view. Some numerical results are also presented to address the analysis of a specific case study.  Guidelines are suggested throughout the paper for the interpretation of  microstructural features, for the use of experimental results in the damage analysis, for best manufacturing practices  and finally for repair strategies.  The described methodologies and the reported results denote a rigorous approach to analysis.

The document is classified as a review paper, but this classification is questionable.  A review paper should be based on a very in-depth and detailed analysis of the literature and not only focused on EPRI’s experience.   The reported results are mainly produced by EPRI experiments. The EPRI experience is mentioned In different parts of the manuscript, thus giving the paper the character of synthesis of internal experiments and of proposal of guidelines for manufacturing, characterization and repair of joints.  For this reason, I would suggest changing the title of the paper, since the current one does not properly describe the content of the manuscript.

The paper deserves to be published in the Journal, creep strength tests on ferritic steel welded joints and the damage models proposed, as well as the manufacturing and repair guidelines are of interest to the scientific and technical community. The paper must undergo to major revisions.

Some suggestions for improvement are listed beliow:

1) the title of the paper must contain a reference to EPRI research and to the proposal of manufacturing, characterization and repair guidelines

2) the introduction must give a more complete description of the results published in the field by other researchers, not only focused on the EPRI experiments.  The abuse of internal reports, as listed in the references, should be avoided, as this material is hardly available for the reader.

3) the results of numerical analyses are of little interest if an in-depth and detailed description of the modeling approach is not provided (see for example fig. 16)

4) other minor points:

a) the promiscuous use of units is not recommended, the best choice would be to present all the data in the SI units, but also the use of both SI and American standard units is acceptable. What is not acceptable is that some results are presented with metric units and other with American units. See for example fig. 2, fig. 8, fig. 13.

b) some typing errors can be found throughout the text, please check the manuscript carefully

Author Response

Some suggestions for improvement are listed beliow:

1) the title of the paper must contain a reference to EPRI research and to the proposal of manufacturing, characterization and repair guidelines. THE TITLE CAN BE MODIFIED TO SAY "EPRI RECOMMENDATIONS FOR Manufacture and Performance of Welds in Creep Strength Enhanced Ferritic Steels" if that is what you would like to see

2) the introduction must give a more complete description of the results published in the field by other researchers, not only focused on the EPRI experiments.  The abuse of internal reports, as listed in the references, should be avoided, as this material is hardly available for the reader. THIS IS NOT THE CASE ALL EPRI REPORTS ARE AVAILABLE TO ALL RESEARCHERS. 

3) the results of numerical analyses are of little interest if an in-depth and detailed description of the modeling approach is not provided (see for example fig. 16). THE BACKGROUND FOR THE WORK SHOWN IN THE FIGURES IS DESCRIBED IN THE REPORT IN SECTION 4, the methods used are also supported by references

4) other minor points:

a) the promiscuous use of units is not recommended, the best choice would be to present all the data in the SI units, but also the use of both SI and American standard units is acceptable. What is not acceptable is that some results are presented with metric units and other with American units. See for example fig. 2, fig. 8, fig. 13. IF THE REVISED MANUSCRIPT IS ACCEPTED THEN THE UNITS WILL BE GIVEN AS SI UNITS

b) some typing errors can be found throughout the text, please check the manuscript carefully, MUCH OF THE MANUSCRIPT HAS BEEN REWRITTEN TO OVERCOME A COMMENT FROM THE EDITOR REGARDING USE OF TEXT FROM PRIOR PAPERS, FINAL EDITS WILL BE PERFORMED IF ERRORS ARE PRESENT. IN MOST PUBLICATIONS THIS TAKES PLACE AT A PROOF STAGE

Reviewer 2 Report

The manuscript: 'Manufacture and Performance of Welds in Creep Strength Enhanced Ferritic Steels' is nicely articulated. I can recommend the manuscript for publication. However, there are a couple of minor comments that need to be rectified before it may be accepted for publication.

- Fig. 1 - some of the scale bars are not readable

- Fig. 5, 11 - Scale bar is missing

Author Response

The manuscript: 'Manufacture and Performance of Welds in Creep Strength Enhanced Ferritic Steels' is nicely articulated. I can recommend the manuscript for publication. However, there are a couple of minor comments that need to be rectified before it may be accepted for publication.

- Fig. 1 - some of the scale bars are not readable. THIS WILL BE CORRECTED IF THE REVISED MANUSCRIPT IS ACCEPTED

- Fig. 5, 11 - Scale bar is missing THIS WILL BE CORRECTED IF THE REVISED MANUSCRIPT IS ACCEPTED

Reviewer 3 Report

This review deals with creep strength of welded component made of enhanced ferritic steels. Authors presented weld manufacture, detailed macro- and microstructure of welded joints and CDM constitutive model for Grade 91 steel. In my opinion section 5 (Advanced Weld Repair Technologies) is the most valuable part of this paper. However it must be strengthened by additional information about laser beam methods, included Direct Deposition 3D printing as very promising, high-tech repair technology.  Summarizing, the review is written in logical way and in good English. Moreover It can be useful for industry community but strongly requires additional information for suitable novelty and science soundness to publishing in Materials journal.

Author Response

Response

This referee makes several positive comments and then ask for the paper to be amended to include laser beam methods and Direct Deposit 3D printing. The paper describes the manufacture and use of steels which are very widely used in power and petrochemical plant. In particular these components operate for long times at high temperatures and pressures. The methods recommended by the referee cannot be used to make any of the components for such applications. Thus, although these methods are relevant for other industries they are not relevant for the paper under consideration

Reviewer 4 Report

The article is well written and interesting from a practical point of view. I would suggest the article for publication. Before final publication, a few comments that the authors could take into account:

Scale bars are not readable in (Figure 1, 6), and some scale bars are missing (Figure 5A). Please revise. 

Figure 2, scale bar is 0.010 in, and unit used in figure caption is um. Measurement system should be consistent throughout the manuscript. 

Figure 13 and Figure 14: the authors use ruler without indicating the measurement system whether is based on inches or millimeters.

References: The reference format could be made more uniform and consistent. Some “pages” are abbreviation “pp”, some are not. Please revise and make sure all bibliographical details, numbering in the list of references must be consistent, complete and accurate.

Author Response

Responses to Comments by reviewer 3 which were inadvertently directed to reviewer 4

This referee makes several positive comments and then ask for the paper to be amended to include laser beam methods and Direct Deposit 3D printing. The paper describes the manufacture and use of steels which are very widely used in power and petrochemical plant. In particular these components operate for long times at high temperatures and pressures. The methods recommended by the referee cannot be used to make any of the components for such applications. Thus, although these methods are relevant for other industries they are not relevant for the paper under consideration

FOLLOWING ARE CORRECT RESPONSES TO REVIEWER 4

The article is well written and interesting from a practical point of view. I would suggest the article for publication. Before final publication, a few comments that the authors could take into account:

Scale bars are not readable in (Figure 1, 6), and some scale bars are missing (Figure 5A). Please revise. AS NOTED THESE ARE VALID COMMENTS AND WILL BE CORRECTED IF THE MANUSCRIPT IS ACCEPTED

Figure 2, scale bar is 0.010 in, and unit used in figure caption is um. Measurement system should be consistent throughout the manuscript. AS NOTED THESE ARE VALID COMMENTS AND WILL BE CORRECTED IF THE MANUSCRIPT IS ACCEPTED

Figure 13 and Figure 14: the authors use ruler without indicating the measurement system whether is based on inches or millimeters.AS NOTED THESE ARE VALID COMMENTS AND WILL BE CORRECTED IF THE MANUSCRIPT IS ACCEPTED

References: The reference format could be made more uniform and consistent. Some “pages” are abbreviation “pp”, some are not. Please revise and make sure all bibliographical details, numbering in the list of references must be consistent, complete and accurate.AS NOTED THESE ARE VALID COMMENTS AND WILL BE CORRECTED IF THE MANUSCRIPT IS ACCEPTED

Round 2

Reviewer 1 Report

accept in the present form

Author Response

Reviewer 1 does not ask for edits and recommends accept

Reviewer 3 Report

Dear Authors,

You are absolutely right. I suggested Direct Deposition 3D printing as new generation repair process for higher novelty and citation impact of your manuscript. That, in fact, actually DD 3D are non-commercial methods. However extensive experimental over world works for using DD repair process for power industry components are conducting. In my opinion it would be worth to notice but decision is yours.

Author Response

This reviewer does not require any changes and states that the decision for including 3D printing is mine. As indicated earlier manufacturing methods for components in Power Boilers must be approved by relevant Codes and Standards. Speculation regarding methods not yet approved does not provide benefit since such methods cannot yet be used for real parts.